# Prognostic Impact of Sleep Patterns and Related-Drugs in Patients with Heart Failure

**DOI:** 10.3390/jcm10225387

**Published:** 2021-11-18

**Authors:** François Bughin, Isabelle Jaussent, Bronia Ayoub, Sylvain Aguilhon, Nicolas Chapet, Sonia Soltani, Jacques Mercier, Yves Dauvilliers, François Roubille

**Affiliations:** 1PhyMedExp, INSERM, CNRS, CHRU, University of Montpellier, 34295 Montpellier, France; f-bughin@chu-montpellier.fr (F.B.); b-ayoub@chu-montpellier.fr (B.A.); j-mercier@chu-montpellier.fr (J.M.); 2INM, INSERM, Université de Montpellier, 34295 Montpellier, France; isabelle.jaussent@inserm.fr; 3Cardiology Department, CHU de Montpellier, 34295 Montpellier, France; s-aguilhon@chu-montpellier.fr (S.A.); s-soltani@chu-montpellier.fr (S.S.); 4Clinical Pharmacy Department, CHU de Montpellier, University of Montpellier, 34295 Montpellier, France; n-chapet@chu-montpellier.fr; 5Unité du Sommeil, Service de Neurologie, Centre National de Référence pour la Narcolepsie, CHU Montpellier, Hôpital Gui-de-Chauliac, 34295 Montpellier, France; y-dauvilliers@chu-montpellier.fr

**Keywords:** heart failure, sleep disturbances, sleep patterns, central nervous system drugs, sleep quality

## Abstract

Sleep disturbances are frequent among patients with heart failure (HF). We hypothesized that self-reported sleep disturbances are associated with a poor prognosis in patients with HF. A longitudinal study of 119 patients with HF was carried out to assess the association between sleep disturbances and the occurrence of major cardiovascular events (MACE). All patients with HF completed self-administered questionnaires on sleepiness, fatigue, insomnia, quality of sleep, sleep patterns, anxiety and depressive symptoms, and central nervous system (CNS) drugs intake. Patients were followed for a median of 888 days. Cox models were used to estimate the risk of MACE associated with baseline sleep characteristics. After adjustment for age, the risk of a future MACE increased with CNS drugs intake, sleep quality and insomnia scores as well with increased sleep latency, decreased sleep efficiency and total sleep time. However, after adjustment for left ventricular ejection fraction and hypercholesterolemia the HR failed to be significant except for CNS drugs and total sleep time. CNS drugs intake and decreased total sleep time were independently associated with an increased risk of MACE in patients with HF. Routine assessment of self-reported sleep disturbances should be considered to prevent the natural progression of HF.

## 1. Introduction

Heart failure (HF) is a common clinical syndrome characterized by inability of the ventricle to fill with or eject blood. HF is a major public health problem with a constantly increasing prevalence [1], a loss of quality of life, and a high mortality rate [2]. Major adverse cardiac events (MACE) are frequent in the history of HF and often lead to hospitalization which is an important marker for poor prognosis [3].

Patients with HF often have several non-cardiac comorbidities which are associated with an impaired quality of life and a poorer prognosis [4]. Identifying and treating the factors contributing to the progression of HF are therefore essential goals for an optimized management. Sleep disturbances are frequent with 60–75% of patients with HF complaining of poor sleep quality, more than 50% of insomnia [5,6] and 35% using regularly hypnotics [7]. The frequency of restless legs syndrome (RLS) varies between 4% and 40% depending on the studies [8,9]. Sleep-disordered breathing are also common (25–66%) in HF patients [10]. The natures and causes of sleep disorders are numerous and often associated in patients with HF, together with the potential for side effects of CNS drugs [11]. Patients with HF and poor sleep quality have increased deterioration in quality of life, alertness, and mood with a higher level of depressive symptoms [12]. A greater morbidity and mortality is also observed in patients with HF and poor quality sleep [12], sleep breathing disorders [13] or taking benzodiazepines [6]. However, most of these studies were limited by short-term follow-up or the use of simple questions instead of validated questionnaires to assess sleep disturbances in this population.

Based on the previous reported studies, we hypothesized that sleep disturbances are linked to a poorer long-term cardiovascular prognosis in patients with HF. Here we aimed to evaluate the association between baseline self-reported sleep characteristics using a comprehensive battery of validated questionnaires and CNS drug intake in 119 patients with HF over a period of 888 days.

## 2. Materials and Methods

### 2.1. Study Design and Population

This study was a longitudinal study on the occurrence of non-fatal and fatal MACE in patients with HF. All participants were admitted to the Cardiology Department, University Hospital Center of Montpellier. The study was approved by the local ethics committee (Institutional Review Board, Montpellier University Hospital, IRB-MTP_2020_05_202000496) (ClinicalTrials.gov identifier NCT04653883).

Among a population of almost 300 patients with HF, 119 patients (median age of 69 years (range = 44–89), 29.41% of women) were included with completed data of the questionnaire. These patients were followed in the Cardiology Department of the Montpellier University Hospital and diagnosed with HF according to the 2016 criteria of the European Society of Cardiology [14]. All patients filled in the questionnaires upon admission to a cardiology day hospital or outpatient clinic.

Patients were recruited between October 2016 and February 2020 and were followed up to the first non-fatal cardiovascular event, death or were censored on 1 March 2020, whichever came first.

### 2.2. Measures

Cardiac pathology, treatment and clinical data were collected on the patient’s computerized medical record by the physicians. The cardiovascular and CNS drugs assessments were carried out by patients and physicians.

A self-administered questionnaire on sleep habits was completed by all patients in order to collect bedtime, waketime, sleep latency, total sleep time (TST), and sleep efficiency (i.e., ratio of the TST compared to the time in bed). RLS was defined using the International RLS Study Group (IRLSSG) criteria. Standardized questions addressed the presence of the four minimal diagnostic criteria of the IRLSSG: (1) Do you feel or have you ever felt an irresistible urge to move your legs? (2) If you feel or you have ever felt an irresistible urge to move your legs, does it begin or become worse during periods of rest or inactivity, such as sitting or lying down? (3) If you feel or you have ever felt an irresistible urge to move your legs, does it improve, at least partially, by movements such as walking or stretching your legs? (4) If you feel or you have ever felt an irresistible urge to move your legs, does-it begin or become worse during the evening or the night? A positive answer to all four questions was required for a presumed diagnosis of RLS [15].

All patients also completed the following standardised questionnaires:Pittsburgh Sleep Quality Index—PSQI [16]

This questionnaire evaluates the quality of the patient’s sleep during the last month before the visit. It is composed of 19 questions concerning 7 items, each item being scored between 0 and 3. The global score therefore varies from 0 to 21 and a high score corresponds to poor quality sleep. A score greater than or equal to 5 defines poor quality sleep.

2.Epworth Sleepiness Scale-ESS [17]

This questionnaire assesses excessive daytime sleepiness (EDS) through 8 different items or situations common or frequently found in everyday life during which the propensity to fall asleep or sleepiness is measured on a scale from 0 to 3. The score is noted from 0 to 24. A score of >10 defines the presence of excessive daytime sleepiness

3.Chalder Fatigue scale [18]

The 11-item Chalder fatigue scale is divided into two components: one that measures physical fatigue (7 questions) and one that measures mental fatigue (4 questions). A global binary fatigue score of 3 or less represents scores of those who are not fatigued, with scores of 4 or more equating to severe fatigue.

4.Insomnia Severity Scale-ISI [19]

This questionnaire estimates the severity of insomnia symptoms over the past month. It includes seven items scored between 0 and 4 with a total score between 0 and 28. The severity of insomnia symptoms is a function of the total score: not clinically significant between 0 and 7, mild between 8 and 14, moderate between 15 and 21 and severe between 22 and 28.

5.Beck Depression Inventory-BDI [20]

This 21-item questionnaire describes the patient’s mood over the 2 weeks preceding the visit. Each item is scored from 0 to 3 with a total score between 0 and 63. A score between 10 and 18 indicates minor depression, 19 and 29 moderate depression, 30 and 63 severe depression.

6.Berlin Questionnaire [21]

This questionnaire is designed to identify people at risk of OSA. The questionnaire is self-administered and consists of 10 questions in three categories related to the presence and severity of snoring, frequency of daytime sleepiness, and the presence of obesity or hypertension. Patients have a ‘high risk’ of OSA if there are 2 or more categories where the score is positive.

### 2.3. CVD Outcome

Occurrence of MACE included death, hospitalization for heart failure, unplanned revascularization, heart transplantation and implantation of a defibrillator or circulatory assistance.

Mortality data were obtained from the public database of INSEE (Institut national de la statistique et des études économiques). Morbidity data were collected from computerized medical record data. These data were verified by the physicians in charge of these patients.

### 2.4. Statistical Analysis

Participants’ characteristics are presented as median (minimum value; maximum value) for continuous variables, or number and percentages for categorical variables.

Cox proportional hazard models with delayed entry and age of the patients as the time scale were used to estimate hazard ratios (HR) and their CI. Demographic and clinical characteristics associated with risk of future MACE at *p* < 0.05 were included in Cox proportional hazard models to estimate the HR for the relationships between each sleep and mood characteristics and risk of future MACE. In the case of multiple events during follow-up, the first event was considered in the survival analysis. Two multivariate models were successively performed: (1) adjustment for left ventricular ejection fraction (LVEF), and hypercholesterolemia and (2) further adjustment for CNS drugs. When appropriate, the interaction terms were tested using the Wald-x2 test. The log-rank test was used to compare the Kaplan–Meier estimate curves for cardia-event free survival with events being right-censored.

Association between baseline characteristics and CNS drugs intake were compared using the Chi-square and Mann–Whitney U tests. Significance level was set at *p* < 0.05. Analyses were performed using SAS statistical software (version 9.4 SAS Inc., Cary, NC, USA).

## 3. Results

Among the 119 patients, respectively 60.50%, 50.00% and 36.75% had hypercholesterolemia, hypertension and diabetes and 28.21% were obese. (Table 1). A HF with reduced ejection fraction (HFrEF) was diagnosed for 46% of the patients. The primary cause of heart failure was ischemic heart disease in 57% of patients. A total 60.50% of patients used angiotensin-converting enzyme (ACE) inhibitors or angiotensin receptor blockers (ARB), 61.34% beta-blockers and 62.18% diuretics. Further, 24.37% took these 3 drugs together.

At baseline, 63.46% complained of poor sleep quality, 21% reported moderate to severe insomnia symptoms, 28.70% an excessive daytime sleepiness and 10.81% had both insomnia symptoms and EDS. RLS was diagnosed in 5.56% of the patients. 35.09% of patients were at high risk for OSA. Around 30% were short sleeper (<7 h/night), 18% long sleeper (≥9 h/night) and 46% reported a sleep efficiency < 85%. Overall, 36 patients (30%) took central nervous system (CNS) drugs (benzodiazepines, Z-drugs (zolpidem and zopiclone) and antidepressant) (Table 2).

Over a median of 888 days follow-up (5–1470), 37 (31.09%) future MACE were observed, including 21 (17.65%) fatal events, 14 (11.76%) acute cardiac failure and 2 (0.02%) cardiac transplantation.

Each baseline socio-demographic and clinical characteristics of the patients according to the occurrence of MACE events during the follow-up are given in Table 1. The risk of future MACE increased in patients with a lower level of LVEF and with more hypercholesterolemia and in patients taking diuretics. We did not include this parameter in the multivariate model given a risk of overfitting (i.e., diuretics intake was associated with a lower LVEF, 35.00 (15.00–67.00) vs. 43.00 (20.00–70.00), *p* < 0.01).

Association between Sleep Parameters, Mood Disturbances, CNS Drugs Intake and Future MACE

Baseline sleep parameters, mood disturbances and CNS drugs intake as a function of MACE events occurrence during the follow-up are given in Table 2. After adjustment for age, the risk of a future MACE increased with CNS drugs intake, PSQI and ISI scores as well as increased sleep latency and decreased sleep efficiency. We also found an increased occurrence of fatal and non-fatal MACE when total sleep time was reduced by one hour. When potential confounders were entered into the model (Model 1, Table 2), the HR were reduced and failed to be significant except for CNS drugs and total sleep time. Similar trends were found however for ISI and PSQI scores. When further adjusting for CNS drugs, the associations became not significant (Model 2, Table 2).

No interaction was found for future MACE event between CNS drugs intake and sleep disturbances.

Patients with HF who were taking CNS drugs had a lower LVEF (*p* = 0.02) and more hypercholesterolemia (*p* = 0.03). Moreover, they received worse scores for BDI (*p* < 0.01), ISI (*p* = 0.02), PSQI (*p* < 0.01) and fatigue dimension of Chalder fatigue scale (*p* = 0.02). Finally, these patients had a lower sleep efficiency (*p* = 0.04).

The Kaplan–Meier survival curves showed that cardiac event-free survival differed in patients who took CNS drugs vs. not (*p* = 0.001) (Figure 1).

## 4. Discussion

Our study reported that sleep disturbances and CNS drugs intake are associated with a poor prognosis in patients with HF with long term follow-up.

This study emphasizes the importance of a detailed drug history in the population of HF patients. Indeed, the main finding of this study was the higher risk of MACE in patients who were taking CNS drugs. CNS drugs include benzodiazepines, Z-drugs and antidepressants (AD). We have chosen to bring these drugs together in a single entity given that their intake is often combined and their respective effects on the CNS are difficult to differentiate. While the relationship between hypnotic use and long-term mortality is not well established in the general population [22] a recent study found that the use of benzodiazepines was an independent predictor of rehospitalisation for HF patients [6]. The poorer prognosis of patients taking these drugs could be related to their potential side effects such as cognitive decline, daytime sleepiness, impairment in psychomotor performance [23] or respiratory depression [24]. The numerous comorbidities associated with the use of CNS drugs [22] and with patients with HF as confirmed in the present study could increase the risk of mortality and cardiovascular morbidity.

We also found an increased occurrence of fatal and non-fatal MACE when total sleep time was reduced by one hour. Short sleep duration (<7 h) is frequently found in the general population [25] and is associated with all-cause mortality [26] although this link is not found in all studies [27]. Moreover, self-reported short sleep (<6 h) is associated with high risk of developing HF in men with cardiovascular disease [28].

In unadjusted analysis, high sleep latency, poor sleep efficiency, poor sleep quality and insomnia symptoms severity were associated with a poorer prognosis in patients with HF. Sleep latency and sleep efficiency are good indicators of sleep quality [29] and should be evaluated to characterize the sleep of patients with HF. Our results were consistent with previous studies for total sleep time [30] and sleep quality [12,28,31]. However, several important differences exist between these studies and ours. First, these studies were secondary analyses, with no primary objective of exploring sleep problems in patients with HF. Secondly, the sleep assessment was often carried out with a single item or a single questionnaire, and never with a comprehensive battery of sleep questionnaires [12].

We did not find any significant results with factors frequently associated with an increased cardiovascular risk. First, sleepiness is a factor of severity in the presence of sleep breathing disorders and is associated with mortality and CVD morbidity [32,33] but it does not appear to be linked to the prognosis of heart failure patients in this study. We did not find any association with the presence of restless legs syndrome, although studies have shown an increase in mortality due to CVD associated or not with periodic leg movements [34,35,36].

There are some limitations in this study. The assessment of sleep parameters was only self-reported causing possible recall bias and a lack of accuracy in responses. However, we used validated questionnaires, which are simple tools that can be used by a large number of practitioners, to assess for sleepiness, fatigue, insomnia symptoms, RLS, proxy of OSA, quality of sleep, anxiety and depressive symptoms. In contrast, the presence of sleep apnea was not evaluated objectively, as no polysomnography was available for this specific study, and the Berlin questionnaire is not very specific in patients with cardiovascular diseases [37]. Finally, the use of diuretics was associated with the occurrence of MACE in univariate analysis, but we did not include this parameter in the multivariate model, because of the risk of overfitting. Indeed, LVEF was closely linked to the use of cardiological drugs with a lower LVEF in patients taking diuretics or beta blockers. Diuretics are used to control symptoms in patients with HF, especially fluid overload, but without effect on mortality [38]. Moreover, approximately 80% of patients with HF are taking diuretics with frequent changes in dosage [39], which makes their integration in the multivariate model irrelevant.

## 5. Conclusions

Our findings suggest that CNS drugs intake and decreased total sleep time were associated with an increased risk of fatal and non-fatal MACE in patients with HF. These results emphasize the importance of routinely screening for sleep symptoms and CNS drugs intake, especially in patients with severe HF to propose a specific management to improve the quality of life and prevent the natural progression of HF.

## Figures and Tables

**Figure 1 jcm-10-05387-f001:**
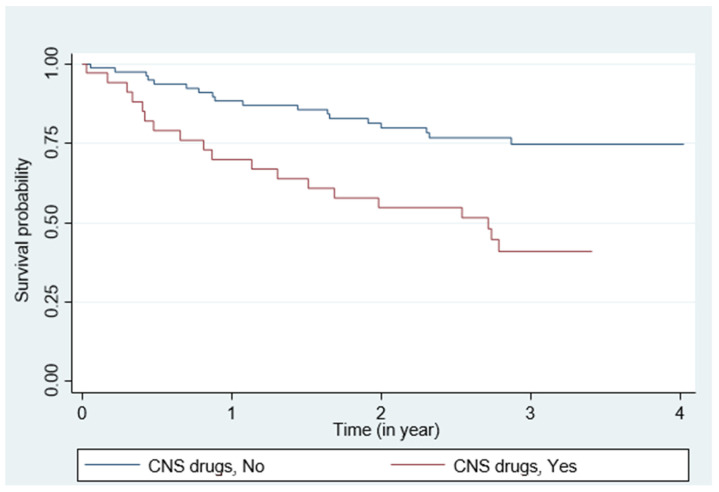
Survival curve of patients with HF according to CNS drugs intake; MACE events.

**Table 1 jcm-10-05387-t001:** Demographic and clinical characteristics of patients according to the occurrence of a CV event during the follow-up.

	Whole Sample		Occurrence of CV Events					
	*n* = 119		No *n* = 82		Yes *n* = 37		Model 0	
Variable	*n*	%	*n*	%	*n*	%	HR (95% CI)	*p*
Sex, Female %	35	29.41	25	30.49	10	27.03	0.79 (0.38; 1.64)	0.531
BMI, kg/m^2 a^	117; 26.56 (18.21; 50.24)		81; 26.17 (18.21; 50.24)		36; 27.43 (18.36; 49.44)		1.03 (0.97; 1.09)	0.352
LVEF, % ^a^ HR for 10% increase	116; 40.00 (15.00; 70.00)		79; 43.00 (18.00; 70.00)		37; 30.00 (15.00; 60.00)		0.74 (0.57; 0.96)	0.021 *
Hypertension, Yes	59	50.00	39	48.15	20	54.05	1.08 (0.57; 2.07)	0.807
Diabetes mellitus, Yes	43	36.75	26	32.10	17	47.22	1.60 (0.83; 3.09)	0.159
Hypercholesterolemia, Yes	72	60.50	43	52.44	29	78.38	2.79 (1.28; 6.11)	0.010 *
Current smoker, Yes	43	36.44	33	40.74	10	27.03	0.57 (0.28; 1.19)	0.133
BDI-score total ^a^	112; 10.50 (0.00; 39.00)		77; 9.00 (0.00; 38.00)		35; 13.00 (3.00; 39.00)		1.03 (0.99; 1.06)	0.139
CFS—Total score ^a^	115; 8.00 (0.00; 14.00)		80; 7.00 (0.00; 14.00)		35; 9.00 (2.00; 14.00)		1.09 (0.99; 1.20)	0.078

* Significant value, ^a^ continuous variables were expressed as number, median (minimal value; maximal value). Model 0: adjusted for age (timescale). Abbreviations: ACEI = angiotensin-converting enzyme inhibitors; ARA = angiotensin II receptor antagonists; BMI = body mass index; OSA = obstructive sleep apnea; BDI = Beck depression inventory; CFS = Chalder fatigue scale; LVEF = left ventricular ejection fraction.

**Table 2 jcm-10-05387-t002:** Self-reported sleep parameters and central nervous system (CNS) drugs intake in patients as function of the occurrence of CV events during the follow-up.

	Whole Sample		Occurrence of CV Events									
			No *n* = 82		Yes *n* = 37		Model 0		Model 1		Model 2	
Variable	*n*	%	*n*	%	*n*	%	HR (95% CI)	*p*	HR (95% CI)	*p*	HR (95% CI)	*p*
CNS drugs, Yes	36	30.25	17	20.73	19	51.35	3.03 (1.59; 5.79)	<0.001 *	2.37 (1.22;4.59)	0.010 *		
ESS score ^a^	115; 7.00 (0.00; 22.00)		80; 6.00 (0.00; 22.00)		35; 9.00 (0.00; 22.00)		1.03 (0.97;1.09)	0.303	1.04 (0.98; 1.11)	0.187	1.04 (0.98; 1.11)	0.234
ESS score, >10	33	28.70	21	26.25	12	34.29	1.31 (0.65; 2.64)	0.448	1.62 (0.80; 3.27)	0.178	1.45 (0.71; 2.93)	0.307
ISI score ^a^	115; 8.00 (0.00; 28.00)		79; 7.00 (0.00; 21.00)		36; 11.00 (1.00; 28.00)		1.07 (1.01; 1.12)	0.012 *	1.05 (1.00; 1.10)	0.048 *	1.03 (0.98; 1.09)	0.197
ISI score, >14	24	20.87	14	17.72	10	27.78	1.52 (0.73; 3.15)	0.261	1.22 (0.58; 2.56)	0.593	0.96 (0.45; 2.07)	0.921
BQ score, ≥2	40	35.09	25	32.05	15	41.67	1.29 (0.67; 2.51)	0.449	1.41 (0.72; 2.74)	0.313	1.05 (0.51; 2.13)	0.903
RLS, ≥4	6	5.56	4	5.56	2	5.56	1.20 (0.29; 5.03)	0.798	1.22 (0.29; 5.09)	0.785	1.43 (0.34; 6.03)	0.630
PSQI score ^a^	104; 6.00 (0.00; 16.00)		71; 6.00 (0.00; 14.00)		33; 7.00 (1.00; 16.00)		1.10 (1.01; 1.20)	0.028 *	1.09 (1.00; 1.19)	0.059	1.05 (0.95; 1.15)	0.362
PSQI score, ≥5	66	63.46	42	59.15	24	72.73	1.29 (0.67; 2.51)	0.268	1.31 (0.60; 2.84)	0.496	0.94 (0.40; 2.18)	0.882
Sleep latency, min ^a^ HR For 15 min Increase	117; 15.00 (0.00; 150.00)		80; 15.00 (0.00; 150.00)		37; 30.00 (0.00; 90.00)		1.11 (0.97; 1.28)	0.127	1.09 (0.94; 1.25)	0.255	1.10 (0.95; 1.28)	0.201
Sleep latency, min												
<30	70	59.83	54	67.50	16	43.24	1	0.033 *	1	0.091	1	0.071
(30–60)	25	21.37	16	20.00	9	24.32	1.54 (0.68; 3.48)		1.47 (0.65; 3.36)		1.13 (0.48; 2.65)	
>60	22	18.80	10	12.50	12	32.43	2.72 (1.28; 5.74)		2.34 (1.09; 5.01)		2.38 (1.10; 5.14)	
TST, hour ^a^ HR For 1 h decrease	119; 7:30 (3:00; 11:45)		82; 7:52 (3:00; 12:00)		37; 7:00 (3:15; 10:15)		0.79 (0.65; 0.98)	0.030 *	0.79 (0.64; 0.97)	0.037 *	0.81 (0.65; 1.01)	0.092
TST, ≤7 h	57	47.90	34	41.46	23	62.16	1.84 (0.95; 3.58)	0.072	1.72 (0.88; 3.36)	0.111	1.41 (0.70; 2.84)	0.333
Efficiency, % ^a^ HR for 10% increase	118; 87.50 (35.29; 100.00)		81; 88.57 (35.29; 100.00)		37; 82.35 (38.24; 100.00)		0.79 (0.65; 0.97)	0.022 *	0.85 (0.69; 1.05)	0.135	0.99 (0.97; 1.01)	0.353
Efficiency ≥85%	63	53.39	45	55.56	18	48.65	0.82 (0.43; 1.57)	0.551	1.02 (0.53; 1.97)	0.947	1.31 (0.66; 2.62)	0.440

* Significant value. ^a^ Continuous variables were expressed as number, median (minimal value; maximal value). Model 0: adjusted for age (timescale), Model 1: adjusted for age (timescale), left. Ventricular ejection fraction (%) and hypercholesterolemia. Model 2: adjusted for covariates in Model 1 plus CNS drugs intake. Abbreviations: BQ = Berlin questionnaire, ESS = Epworth severity scale, HR = hazard ratio, ISI = insomnia severity index, PSQI = Pittsburgh sleep quality index. TST = total sleep time, RLS = restless legs syndrome.

## Data Availability

Data would be provided under reasonable request.

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
