# Peer review of "Prognostic Impact of Sleep Patterns and Related-Drugs in Patients with Heart Failure"

_jcm, 2021, doi:10.3390/jcm10225387_

Round 1

Reviewer 1 Report

In this work, Bughin et al. developed a statistical study in 119 patients with heart failure showing that the intake of central nervous system drugs and decreased total sleep time are associated with an increased risk of suffering major cardiovascular events. Overall the paper is well organized and written, but several issues have to be addressed before publication.

Comments:

  1. Line 151 states that 60.5% of the patients had dyslipidemia, but this % corresponds to the patients with hypercholesterolemia in Table 1.
  2. Line 136 indicates that the numeric variables are presented as mean and standard deviation, but the values in the tables are expressed as the minimum, the maximum, and the median.
  3. In the result section, Table 2 should be called somewhere between lines 154 and 166.
  4. All the p-values need to follow the same format, three decimals, and the abbreviation (p<0.001) for small values.
  5. To facilitate the table reading, the significant values need to be indicated with an asterisk or any other annotation.
  6. Remove the % symbol from the HR statistic in Table 1.
  7. Lines 165 and 166 mention that the risk of future MACE increases in patients with low levels of LVEF and more hypercholesterolemia, but the authors ignore the use of diuretics when the HR value is 6.73. This value suggests that this factor may also be relevant to the occurrence of MACE. Why do the authors dismiss this variable?
  8. LVEF variable shows a p-value=0.02 in Table 1, but the confidence interval (HR) contains the value 1 ([0.95;1.00]). This is impossible if the variable is significant.
  9. This referee still does not understand why the authors did not include the variable “diuretics” in the Cox multivariant model. Its HR value in model 0 is higher than the HR values from the variables studied in models 1 and 2.
  10. Did the authors adjust models 1 and 2 by age since that was the case for model 0? The authors should do this if they want to compare models 0, 1, and 2.
  11. In Table 2, did the authors show a multivariate regression analysis including 15 independent variables and some other adjustment variables? or did they calculate the Cox regressions for each independent variable? If we are in the first case, the authors should not include numerical and categorical variables (ESS or IS) in the same regression analysis since 1) it may reduce the efficiency of the model and 2) the sample size of the study does not meet 1996 Peduzzi’s criteria. In the second case, it would be very interesting to check the Mace set of risk factors by including the significant variables from models 0,1, and 2 in the Cox multivariant model.

Author Response

Point 1: Line 151 states that 60.5% of the patients had dyslipidemia, but this % corresponds to the patients with hypercholesterolemia in Table 1.

Response 1: Responses: we agree with the reviewer. We modified the sentence accordingly:

“Among the 119 patients, respectively 60.50%, 50.00% and 36.75% had hypercholesterolemia, hypertension and diabetes and 28.21% were obese”

Point 2: Line 136 indicates that the numeric variables are presented as mean and standard deviation, but the values in the tables are expressed as the minimum, the maximum, and the median.

Response 2: We have expressed continuous variables as median, minimum and maximum values. We modified the sentence accordingly:

“Participants’ characteristics are presented as median [minimum value; maximum value] for continuous variables, or number and percentages for categorical variables.”

Point 3: In the result section, Table 2 should be called somewhere between lines 154 and 166.

Response3 : We referred the Table 2 in the first results section accordingly:

“At baseline, 63.46% complained of poor sleep quality, 21% reported moderate to severe insomnia symptoms, 28.70% an excessive daytime sleepiness and 10.81% had both insomnia symptoms and EDS. RLS was diagnosed in 5.56% of the patients. 35.09% of patients were at high risk for OSA. Around 30% were short sleeper (<7 hours/night), 18% long sleeper (≥9 hours/night) and 46% reported a sleep efficiency <85%. Overall, 36 patients (30%) took central nervous system (CNS) drugs (benzodiazepines, Z-drugs (zolpidem and zopiclone) and antidepressant) (Table 2).”

Point 4: All the p-values need to follow the same format, three decimals, and the abbreviation (p<0.001) for small values.

Response 4:  We modified the tables. All the p-values are in the same format in the revised manuscript.

Point 5: To facilitate the table reading, the significant values need to be indicated with an asterisk or any other annotation.

Response 5: we added an asterisk “*” in the tables to highlight significant results.

Point 6: Remove the % symbol from the HR statistic in Table 1.

Response 6: We remove the “%”

Point 7: Lines 165 and 166 mention that the risk of future MACE increases in patients with low levels of LVEF and more hypercholesterolemia, but the authors ignore the use of diuretics when the HR value is 6.73. This value suggests that this factor may also be relevant to the occurrence of MACE. Why do the authors dismiss this variable?

Response 7: We thank the reviewer for this comment.

We decided in this revised version to not include the different cardiological drugs in the table. We also decided to not include the variable “diuretics” in the Cox multivariant model for 2 reasons. First, diuretic use (as well as beta blockers) was related to LVEF with a lower LVEF in patients taking diuretics (p<0.01), with same results for those taking beta blockers (p=0.01).  Diuretics are often used to control symptoms in patients with HF, especially fluid overload, but without effect on mortality [see new reference 38]. Moreover, approximately 80% of patients with HF are taking diuretics with frequent changes in dosage [ see new reference 39]], which makes their integration in the multivariate model irrelevant. Second, in order to respect the Peduzzi criteria[1], we are limited in the number of adjustment variables due to the number of patients included. For these reasons, we did not integrate the use of diuretics in the model in order to avoid an overfitting. We modified the results in adding these data in the text, and explained this choice in the discussion accordingly.

Point 8: LVEF variable shows a p-value=0.02 in Table 1, but the confidence interval (HR) contains the value 1 ([0.95;1.00]). This is impossible if the variable is significant.

Response 8: We agree with the reviewer. LVEF was associated with the occurrence of CV events at p=0.021. The HR and its confidence intervals have been rounded to two decimals after the point, resulting in a confidence interval bound to 1.00 instead of 0.996. To avoid misunderstanding, we expressed the results of the HR for 10 units increase instead of 1 unit increase with a HR=0.74 95%CI=[0.57;0.96] for 10% increase. We included this result in the Table 1.

Point 9: This referee still does not understand why the authors did not include the variable “diuretics” in the Cox multivariant model. Its HR value in model 0 is higher than the HR values from the variables studied in models 1 and 2.

Response 9: We did not include the variable “diuretics” in the Cox multivariant model due to the risk of over-adjustment as explained in point 7 above.

Point 10: Did the authors adjust models 1 and 2 by age since that was the case for model 0? The authors should do this if they want to compare models 0, 1, and 2.

Response 10: All the models were adjusted for age. We used Cox proportional hazard models with delayed entry and age of the patients as the time scale. This method gives better adjustment for age and is therefore preferable for a sample of elderly individuals over the standard model that uses study time as the time scale, because the covariates are strongly associated with age. We added age (timescale) in the legend of the model 1 Table 2 :

“Model 1: adjusted for age (timescale), left Ventricular Ejection Fraction (%) and hypercholesterolemia.”

Point 11: In Table 2, did the authors show a multivariate regression analysis including 15 independent variables and some other adjustment variables? or did they calculate the Cox regressions for each independent variable? If we are in the first case, the authors should not include numerical and categorical variables (ESS or IS) in the same regression analysis since 1) it may reduce the efficiency of the model and 2) the sample size of the study does not meet 1996 Peduzzi’s criteria. In the second case, it would be very interesting to check the Mace set of risk factors by including the significant variables from models 0,1, and 2 in the Cox multivariant model.

Response 11: In table 2, we examined the association between each self- reported parameters plus CNS drugs with the occurrence of CV events as an independent manner. We modified the statistical section to avoid misunderstanding accordingly:

“Cox proportional hazard models with delayed entry and age of the patients as the time scale were used to estimate hazard ratios (HR) and their CI. Demographic and clinical characteristics associated with risk of future MACE at p<0.05 were included in Cox proportional hazard models to estimate the HR for the relationships between each sleep and mood characteristics and risk of future MACE.”

We agree with the reviewer that it could be nice to implement in a same model parameters associated with the occurrence of CV events with the adjustment variables. However, we have only 37 CV events in our sample. If we applied the rule of thumb that cox models should be used with a minimum of 10 events per adjusted variable (Peduzzi et al., J Clin Epidemiol, 1995), we cannot implemented such models, the number of outcome events being “too small”.

[1] Peduzzi et al., « Importance of Events per Independent Variable in Proportional Hazards Regression Analysis II. Accuracy and Precision of Regression Estimates ».

Reviewer 2 Report

In this article the authors test the hypothesis that self-reported sleep disturbances are associated with a poor prognosis in patients with heart failure. I have some comments:

Results of Cox proportional hazard models need to be detailed. It is not clear, for example, the general characteristics of the model. I am guessing that the results are presented in Table 1, but it is not clearly stated in the text. Clarify why the CNS drugs were not used in the overall model?

Author Response

In this article the authors test the hypothesis that self-reported sleep disturbances are associated with a poor prognosis in patients with heart failure. I have some comments:

Results of Cox proportional hazard models need to be detailed. It is not clear, for example, the general characteristics of the model. I am guessing that the results are presented in Table 1, but it is not clearly stated in the text. Clarify why the CNS drugs were not used in the overall model?

In table 1, we examined the associations of each demographic and clinical characteristics with the outcome variable (occurrence of a CV event during the follow-up). Demographic and clinical characteristics associated with the occurrence of a CV event at p<0.05 were introduced in several multivariate models as adjustment variables to examine the associations of each sleep, mood parameters, and  CNS drugs with the occurrence of a CV event (table 2). Each sleep, mood parameters and CNS drugs were thus examined with the occurrence of a CV event 1) after adjustment for age (model 0), 2) after adjustment for age, left Ventricular Ejection Fraction (%) and hypercholesterolemia 3) after adjustment for age, left Ventricular Ejection Fraction (%) and hypercholesterolemia and CNS drugs intake (Table 2). We modified the statistical sections  in the revised version.

Reviewer 3 Report

The emphasis of a detailed sleep drug history and sleepy quality assessment in patients with heart failure in the article is important.  However, there is no clear description about patient screening, like how these 119 patients were screened out? What is the proportion of patients participating in all hospitalized patients? Another question is that is there any confounding between sleep quality and severity of heart failure, like NYHA class.

Author Response

The emphasis of a detailed sleep drug history and sleepy quality assessment in patients with heart failure in the article is important.  However, there is no clear description about patient screening, like how these 119 patients were screened out? What is the proportion of patients participating in all hospitalized patients? Another question is that is there any confounding between sleep quality and severity of heart failure, like NYHA class.

We offered the questionnaire to nearly 300 patients diagnosed with HF upon admission to a cardiology day hospital or outpatient in the cardiology department (e.g. 100 new patients per year) . Only patients who correctly completed the questionnaires were included. We modified the Method section accordingly.

Regarding the question on the relationship between the sleep quality and the severity of heart failure, we preferred for this study to include the data from the LVEF instead of the ones from the NYHA classification. The NYHA classification was not always reported in the medical record and not often at the same time as the completion of sleep questionnaire. We have preferred to avoid reporting these uncertain data in the present article.

Round 2

Reviewer 1 Report

The authors have done a convincing job in addressing my previous comments and editing the manuscript so I am ready to recommend acceptance.